# The Cellular Response to Complex DNA Damage Induced by Ionising Radiation

**DOI:** 10.3390/ijms24054920

**Published:** 2023-03-03

**Authors:** Beth Wilkinson, Mark A. Hill, Jason L. Parsons

**Affiliations:** 1Department of Molecular and Clinical Cancer Medicine, University of Liverpool, 6 West Derby Street, Liverpool L7 8TX, UK; 2MRC Oxford Institute for Radiation Oncology, University of Oxford, Old Road Campus Research Building, Roosevelt Drive, Oxford OX3 7DQ, UK; 3Institute of Cancer and Genomic Sciences, University of Birmingham, Edgbaston, Birmingham B15 2TT, UK

**Keywords:** carbon ions, complex DNA damage, DNA repair, ionising radiation, linear energy transfer, proton beam therapy

## Abstract

Radiotherapy (ionising radiation; IR) is utilised in the treatment of ~50% of all human cancers, and where the therapeutic effect is largely achieved through DNA damage induction. In particular, complex DNA damage (CDD) containing two or more lesions within one to two helical turns of the DNA is a signature of IR and contributes significantly to the cell killing effects due to the difficult nature of its repair by the cellular DNA repair machinery. The levels and complexity of CDD increase with increasing ionisation density (linear energy transfer, LET) of the IR, such that photon (X-ray) radiotherapy is deemed low-LET whereas some particle ions (such as carbon ions) are high-LET radiotherapy. Despite this knowledge, there are challenges in the detection and quantitative measurement of IR-induced CDD in cells and tissues. Furthermore, there are biological uncertainties with the specific DNA repair proteins and pathways, including components of DNA single and double strand break mechanisms, that are engaged in CDD repair, which very much depends on the radiation type and associated LET. However, there are promising signs that advancements are being made in these areas and which will enhance our understanding of the cellular response to CDD induced by IR. There is also evidence that targeting CDD repair, particularly through inhibitors against selected DNA repair enzymes, can exacerbate the impact of higher LET, which could be explored further in a translational context.

## 1. Introduction

There are an estimated 19 million new cases of cancer diagnosed, and 10 million cancer deaths each year [1]. As many as 50% of all cancer patients will receive some form of radiotherapy as part of their treatment, either alone or in combination with surgery, chemotherapy or immunotherapy [2]. Radiotherapy utilises ionising radiation (IR), generally low linear energy transfer (LET) X-rays (photons) to treat the tumour, but which can cause acute and long-term side effects due to the irradiation of the surrounding tissues and organs at risk. LET refers to the amount of energy loss by an ionising particle per unit distance travelled and relates the density of ionisation events along the radiation track [3]. In contrast, the use of particle ions, such as proton beam therapy (PBT) and carbon ion therapy (CIRT) has significant advantages over photon radiotherapy, since the entrance dose is lower, radiation can be delivered within a well-defined region (called the Bragg peak) that can be targeted to the tumour, and then there is a low exit dose (Figure 1A) [4]. Beams of differing initial energies can also be combined to produce a spread-out Bragg peak (SOBP) that allows the specific targeting of larger tumour volumes (Figure 1B,C). Despite the first clinical uses of PBT and CIRT being in the 1950s and 1970s, respectively, only recently has this technology become more increasingly and commonly used [5]. One of the major factors for this is the biological and clinical uncertainty due to the increases in LET at the Bragg peak, and particularly the distal fall-off. Indeed, a relative biological effectiveness (RBE) for PBT of 1.1 is used clinically which has been highly debated [6,7], whereas the RBE of CIRT is as high as 3–4 in the Bragg peak region due to the increased ionisation density associated with the significantly higher LET. The use of CIRT also offers radiobiological advantages, for example, due to the high-LET, CIRT is less sensitive to variations in oxygen concentration, cell cycle distribution and radiosensitivity of the cells within the tumour. Unlike for protons, the variation of RBE along the carbon ion beam is routinely accounted for using radiobiological models (Figure 1C). The interaction of IR with biological and cellular components can occur either through direct ionisation or excitation of the macromolecules such as DNA, or indirectly largely through the radiolysis of water to create reactive oxygen species, most notably hydroxyl radicals, which can subsequently react with nearby macromolecules. Due to the highly reactive environment within the cell, the diffusion distance of these radicals is <10 nm, so damage is produced in close proximity to the original radiation track [3]. Nevertheless, and despite the radiation modality, the critical cellular target that drives tumour cell killing is DNA.

In terms of ionisation density, X-rays/γ-rays along with high energy PBT are relatively sparely ionising and therefore low-LET, whereas carbon ions are densely ionising and high-LET. This is important as changes in LET can impact on the levels and complexity of the DNA damage induced. For low-LET radiation, approximately 1000 DNA single strand breaks (SSBs), 40 DNA double strand breaks (DSBs) and 1300 DNA base lesions are generated per 1 Gy dose [8]. These typical types of DNA damage are largely generated through an indirect mechanism, where the probability of forming a damage lesion is sensitive to the oxygen concentration. This contrasts with higher LET IR, where the higher density of ionisation events along the radiation track can ultimately result in clusters of lesions (which can include both direct and indirect damage) within a few base pairs. As a result, the biological effectiveness of these clustered lesions is typically less dependent on oxygen, as even in the absence of oxygen and associated reduction in indirect lesions, these sites of damage still contain multiple lesions which are difficult to repair. Complex DNA damage (CDD) is defined as two or more lesions within one to two helical turns of the DNA (which include DSBs), however, it has proven challenging to determine the actual frequency and complexity of this type of damage generated following IR, especially as a spectrum of CDD is produced. Mathematical modelling has revealed that even for low-LET radiation, up to 50% of DSBs can have an additional strand break and/or base damage in close proximity, and are therefore deemed as complex DSBs [9,10]. Nevertheless, the higher the LET, the greater the frequency and complexity of CDD induced. For example, low-energy (1 MeV) protons have been estimated to generate ~80% DSBs that are complex, whilst this can increase to >90% with high-LET α-particles [11]. Given the structurally and chemically complex nature of the damage, CDD represents a challenge to the cellular DNA repair machinery compared with that of isolated DNA lesions. For instance, while pulse field gel electrophoresis has shown that the majority of low-LET DSB are repaired with ~20 min half time [12] and similarly, data from neutral comet assays reveal that the majority of DSBs are repaired between 1 and 2 h [13,14], complex DSBs associated with high-LET radiation repair at a much slower rate. These can therefore persist for several hours post-irradiation and the reduced repairability of the damage can drive IR-induced cell death.

Here, we provide an up-to-date review of the cellular and biological responses to CDD, particularly in the context of IR with increasing LET. We also highlight some of the techniques currently used to measure CDD, and opportunities to target the enzymes and pathways involved in the repair of CDD to enhance the biological effects of IR.

## 2. Cellular DNA Damage Response (DDR)

DNA is continually subject to stress induced endogenously through the formation of reactive oxygen species (ROS), as well as via exogenous sources such as IR. The DNA damage response (DDR) is a sophisticated signalling network that is used in cells to detect and repair a range of different DNA lesions. In terms of IR, the major types of DNA lesions induced are SSBs, base damage (sites of base loss/AP sites, or oxidised DNA bases) and CDD, which includes simple and complex DSBs, as well as non-DSB complexes. In fact, what makes IR unique from endogenous damage and its biological effectiveness is its efficiency at inducing CDD. Base damage and SSBs are repaired by proteins of the BER pathway (Figure 2A) [15,16]. Repair of damaged DNA bases is initiated by one of eleven damage-specific DNA glycosylases that excise the lesion, although the major DNA glycosylases that are responsive to oxidative DNA damage include 8-oxoguanine DNA glycosylase (OGG1), endonuclease III homologue (NTH1) and the endonuclease VIII-like 1–3 proteins (NEIL1-3). Once excised, the abasic site generated is recognised and incised by AP endonuclease 1 (APE-1) in the case of OGG1 and NTH1. In contrast, following the bifunctional activity of NEIL1-3, this generates single strand breaks ends that require processing by polynucleotide kinase 3’-phosphatase (PNKP). Subsequently, DNA polymerase β (Pol β) and the complex of X-ray repair cross complementing protein 1-DNA ligase IIIα (XRCC1-Lig IIIα) insert the correct undamaged nucleotide and seal the nick in the DNA, respectively. It should be noted that SSBs generated as intermediates of BER or generated directly, are recognised by poly(ADP-ribose) polymerase 1 (PARP-1), which protects the strand break but also plays a role in the recruitment of downstream repair proteins.

DSBs are usually repaired through either the non-homologous end joining (NHEJ) or homologous recombination (HR) pathways, the latter of which is only active during S/G_2_ phases of the cell cycle when there is a sister chromatid available for repair. NHEJ can be sub-divided into two pathways, namely, classical/canonical NHEJ (cNHEJ; Figure 2B) and alternative NHEJ (aNHEJ; Figure 2C) [17]. However, the starting point for DSB repair is the phosphorylation of the histone variant H2AX (also termed γH2AX) performed by the protein kinases ataxia telangiectasia mutated (ATM) and ataxia telangiectasia and Rad3-related (ATR) enzymes [18]. cNHEJ begins through DSB detection and binding by the Ku70/Ku80 heterodimer. DNA-dependent protein kinase catalytic subunit (DNA-PKcs) and the nuclease Artemis are recruited, nucleotide addition is then completed by DNA polymerases μ or λ, and ligation of the DSB is performed by a complex consisting of X-ray repair cross complementing protein 4 (XRCC4), DNA ligase IV (Lig IV) and XRCC4-like factor (XLF). In aNHEJ, the MRE11-RAD50-NBS1 (MRN) complex in concert with carboxy-terminal binding protein interacting protein (CtIP) resects the DNA ends to generate 3′-overhangs and regions of microhomology. PARP-1 is then recruited along with DNA polymerase θ that performs DNA synthesis, and finally the DNA is ligated through the activity of XRCC1-Lig IIIα or DNA ligase I (Lig I). During the HR pathway of DSB repair [19], and following end resection by the MRN complex, the 3′ single-stranded DNA tail is protected and stabilised by replication protein A (RPA) (Figure 2D). RPA is then replaced with BRCA2/RAD51, which initiates homology search and strand nucleofilament invasion using the sister chromatid. DNA synthesis and ligation generate a Holliday junction, which is then processed by resolvases.

In terms of the repair of CDD, this very much depends on the nature and complexity of the damage, and particularly the LET of the IR being examined. It can be assumed that complex SSBs will largely require proteins involved in BER if the individual DNA lesions are repaired sequentially. However, the close proximity of the lesions making up these non-DSB CDD sites can significantly impair the repair of the individual lesions, increasing their persistence and the probability that they are still present during replication [20]. Complex DSBs, on the other hand, are likely to involve enzymes across all the repair pathways, BER, NHEJ and HR. Current evidence of the precise enzymes and mechanisms involved in the signalling and repair of CDD is covered in more detail in Section 3.3.

## 3. Complex DNA Damage (CDD)

### 3.1. Generation and Biological Consequences of CDD

CDD is a hallmark of damage induced by the track structure of IR, and these sites are notoriously difficult for the cell to repair which is exploited in the use of radiotherapy for the treatment of cancer, and particularly for high-LET radiotherapy such as CIRT. The simplest form of a CDD site is actually a DSB, formed by two SSBs in close proximity. However, as the LET of the IR increases, so does the degree and frequency of complexity of the damage, although it has been shown there are no dramatic increases in the levels of isolated DSBs and SSBs, and in fact, these actually decrease [21,22]. Nevertheless, in terms of quantitative levels of CDD, estimates from mathematical modelling have suggested that up to 40% of SSBs and up to 50% of DSBs generated by low-LET radiation can have additional damage in close proximity, and therefore are deemed CDD [9,10,11]. It has also been shown experimentally following low-LET radiation that the frequency of non-DSB CDD is approximately four times that of the levels of isolated DSBs [23,24]. PBT at low energies (<4 MeV), and with increasing LET, appears to demonstrate increases in the levels of CDD (both complex SSBs and DSBs), whereas with high-LET α-particles, the majority of the DNA DSB damage (>90%) is complex in nature. The increasing ionisation density/LET will give rise to increased complexity of the damage that has reduced repairability compared to isolated DNA lesions that are formed routinely in the cell as a consequence of endogenous metabolism. Therefore, along with DSBs which are considered the most toxic DNA damage, CDD is a major contributor to the biological effects induced by IR, particularly relating to cell lethality in the context of radiotherapy used in cancer treatment.

A number of studies have demonstrated that CDD induced by IR has a longer lifetime in cells compared to more simple DNA lesions. As a few examples, in human monocytes it has been demonstrated using gel electrophoresis that abasic site-associated CDD was evident over a period of 14 days following 5 Gy γ-irradiation compared to DSBs that were completely joined within 1 day [25]. This contrasts greatly with the vast majority of DSB induced by γ-rays that are repaired within 1 h. Interestingly and using a similar methodology, it has been demonstrated that oxidative-associated CDD could be detected in the skin tissue of mice exposed to 12.5 Gy X-ray irradiation 20 weeks after treatment [26]. Our more recent research has shown using an enzyme-modified neutral comet assay that CDD induced by low energy PBT (generated at the Bragg peak) in HeLa cells persists for at least 4 h post-irradiation at a time when all the frank DSBs have been repaired, and which correlates with the enhanced biological effectiveness compared to relatively higher energy protons [13,27]. However, CDD induced by comparatively higher-LET radiation leads to sites of DNA damage that are significantly more challenging to repair. This has been observed in human fibroblasts exposed to Fe ions and examining co-localisation of DNA repair proteins by immunofluorescence microscopy to show that this persisted and was relatively unchanged 24 h post-irradiation [28]. Similar observations of significant CDD persistence have been seen in other studies following Fe ion exposure [29,30]. Additionally, significant delays in the repair of DSBs up to 24–48 h post-irradiation through γH2AX foci analysis have been demonstrated in glioblastoma [31] and salivary gland cells [32] following CIRT versus photon radiation.

In terms of biological consequences, the greater RBE and cell killing effects of high-LET radiation through its increased ionisation density can, not only be correlated with the increased levels and complexity of CDD within the nucleus, but also the increasing correlation of multiple CDD along individual radiation tracks. To this effect, it has been demonstrated that complex DSBs lead to a phenomenon known as chromothripsis, where extensive local chromatin fragmentation results in catastrophic genomic rearrangements. This has been shown indirectly through the use of Chinese hamster ovary cells containing constructs with multiple I-*Sce*I restriction sites generating several DSB sites within a few hundred base pairs [33]. Interestingly, chromatin compaction has also been shown to affect the spacing and quantity of complex DSBs induced by CIRT as viewed using transmission electron microscopy (TEM) [34]. It was observed that larger clusters of DSBs were found localised at condensed heterochromatin, and that some of these CDD sites that remained unrepaired persisted for up to 48 h. Low-LET induced DSBs, in contrast, were repaired within 24 h in both euchromatin and heterochromatin. However, it should be noted that cell cycle phase can affect cellular radiosensitivity. It is established that following low-LET radiation, cells show their highest sensitivity during the G_2_/M phase, and significantly less sensitivity during the S-phase [35]. In contrast, it is unclear whether cell cycle phase has a major influence on the sensitivity of cells to CDD, particularly that are induced by high-LET radiation. Not only can CDD induce cytotoxicity, but there is a long line of evidence largely generated from oligonucleotide substrates and plasmid-based systems transfected into *E. coli*, yeast and mammalian cells that CDD is mutagenic (summarised in [20,36]). This stems again from the inability of the DNA repair machinery to efficiently and accurately repair the DNA damage leading to deletions, base substitutions and insertions.

### 3.2. Measurement of CDD

Measuring CDD induction and the subsequent repair pathways utilised to resolve the DNA damage has not been an easy task experimentally. Monte Carlo based simulations have long been used to predict the frequency and complexity of DNA damage induced within cells in response to IR of increasing LET (for example [37]). However, it is only recently they have started to take into account the complexities of the DNA structure packaged within chromatin [38,39], but there is still work to be done with respect to taking into account the associated cellular environment. Historically, synthetic oligonucleotides containing site-specific DNA damages (such as abasic sites and oxidative base damage) that mimic CDD induced by IR have also been utilised to analyse their repairability by purified DNA repair proteins or cell extracts (summarised in [20,40]). Data from these experiments has provided important molecular insight into the processing of the lesions, and potential hierarchy in terms of excision and repair of the individual damages present within a cluster. However, again, these suffer from being very artificial in nature, and in terms of monitoring their repair, they do not accurately reflect the complex biological processes involved in recognising the CDD in chromatin, and the interactions between chromatin remodelling and DNA repair enzymes mediated through extensive histone and protein post-translational modifications. More recently, the processing of CDD sites by cell extracts has been examined in mononucleosomes constructed using oligonucleotide substrates and purified histones, which at least replicate the basic chromatin structure [41,42]. These experiments have revealed that there is restricted access of the lesions within nucleosomes to DNA repair enzymes, but which very much depends on the position, orientation, but also the type of lesion existing. This, nevertheless, re-enforces the need for chromatin modelling processes to take place in order to stimulate CDD repair. Cytogenetics is another method that has long been used to analyse DNA damage at the chromosomal level, and which can be correlated with the degree of CDD induced by IR of increasing LET. If a DSB fails to repair and is towards the end of a chromosome, this may result in a terminal deletion leading to DNA loss. Alternatively, two DSBs may misrepair in a pairwise fashion, resulting in chromosome aberrations, with the frequency of these increasing with decreasing distance between the breaks. The cytokinesis-block micronucleus (CBMN) assay [43] can be used to quantitate micronuclei generated as a result of DNA damage and ultimately chromosomal fragments that are excluded from the daughter nuclei during mitosis and become enveloped in their own membrane. Similarly, fluorescence in situ hybridisation (FISH) can be used to observe chromosomal rearrangements, although the limitation of these techniques is that CDD formation is inferred through the degree of chromosomal damage (particularly in direct comparison to low-LET radiation) rather than being a direct measurement and analysis of CDD. For low doses of low-LET radiation, the resulting aberrations are often simple (maximum of two breaks in two chromosomes), whereas high-LET radiation predominantly produces complex aberrations (exchanges involving three or more breaks in two or more chromosomes) [3,44]. However, the true complexity of rearrangements is likely to be underestimated due to the limited resolution (~10 Mbp) of these techniques.

It is therefore clear that direct visualisation and quantification of CDD induced by IR in cells is an essential tool in order to further understand the molecular and cellular pathways responsive to this type of damage generated largely by high-LET radiation. Using DNA glycosylases and endonucleases to cleave persistent DNA base damage and abasic sites, along with the separation of the DNA through the use of gel electrophoresis, was one of the first techniques used for the direct detection of CDD [45]. Depending on the specific enzyme used (for example, formamidopyrimidine DNA glycosylase (Fpg) or endonuclease III (Nth)) this can reveal CDD associated with either abasic sites, oxidised purines or oxidised pyrimidines [23]. DNA repair enzymes can also be used in combination with the neutral comet assay to determine the levels of DSB-associated CDD or base damage/SSB in close proximity that subsequently generates a DSB upon cleavage [46,47]. The enzyme-modified comet assay involves embedding cells within agarose on a microscope slide followed by lysis using high concentrations of salt and non-ionic detergent. The DNA is then treated with the appropriate enzymes to incise the unrepaired DNA base damage, and following DNA unwinding and electrophoresis in a relatively neutral pH buffer, a comet tail is formed when the DNA migrates from the nucleoid core. This assay is relatively straightforward to use to determine IR-induced CDD levels, but is particularly useful for analysing the cellular repair kinetics and persistence of the damage post-treatment. For example, the enzyme-modified neutral comet assay has been used to show that HeLa cells irradiated with relatively high-LET PBT generate CDD that was persistent for at least 4 h post-treatment and correlates with an increased RBE, compared to irradiation of cells with low-LET PBT where no significant CDD formation was evident [13]. However, both of these electrophoresis-based techniques only reveal the total levels of CDD and its repair, rather than yielding important information on the degree of complexity and actual site of occurrence of the damage within the DNA.

The phosphorylation of the histone variant H2AX (designated γH2AX) as a surrogate marker of DSBs at the cellular and tissue level has been used routinely since its discovery in 1998 [18]. Individual γH2AX foci indicating the sites of DSB damage and repair can be revealed and quantified using immunofluorescence microscopy, or even combined with flow cytometry [48]. A number of reports have used not only the persistence of γH2AX foci, but also foci size to indicate possible CDD sites, particularly when comparing low versus high-LET radiation. Indeed, in lung epithelial cells it has been demonstrated that γH2AX foci were ~20–50% larger with α-particles and persisted for longer times post-irradiation in a dose-dependent manner compared to X-rays [49]. However, these foci may actually contain multiple individual overlapping foci as a result of the production of closely correlated sites of CDD along the track of individual α-particles. Additionally, super resolution nanoscopy has demonstrated ~25–40% differences in sizes of γH2AX and 53BP1 foci in HeLa cells following PBT and CIRT [50]. However, another approach in order to analyse CDD sites, particularly those associated with DSBs, is through the co-localisation of γH2AX with non-DSB proteins such as OGG1 and APE1 [51]. This has been used to demonstrate that there is increased CDD with increasing dose and LET (from X/γ-rays to α-particles). This approach can at least allow the visualisation and localisation of CDD within regions of the DNA in the nucleus, and could potentially be used to examine those sites present and repaired within euchromatin versus heterochromatin.

In addition to immunofluorescence microscopy, alternative forms of microscopy have been used to analyse CDD sites. Atomic force microscopy (AFM) has been utilised to detect small DNA fragments that are produced following high-LET irradiation of plasmid DNA [52]. This analysis showed that by comparing IR of increasing LET, ranging from low-LET electrons and γ-rays to high-LET beryllium and argon ions, the size of DNA fragments shortened, which reflected the increase in the formation of DSB-associated CDD. More recently, a methodology utilising purified genomic DNA in combination with AFM has been used to evaluate the yield and complexity of CDD in cells irradiated with X-rays and Fe ion beams [53]. This involves the use of DNA glycosylases (Nth and OGG1) to generate abasic sites which are then labelled with an aldehyde reactive probe (ARP) containing a biotin tag, and following DNA digestion and streptavidin magnetic bead capture, the DNA damage can be observed using AFM. Analysis using this approach demonstrated increased base damage-associated CDD relative to isolated DNA damage following Fe ions versus X-rays, however intriguingly, there appeared to be no difference in the levels of complex DSBs between the two different IR sources. Nevertheless, it was revealed that the CDD induced persisted for at least 18 h post-irradiation, consistent with other reports and more commonly used methodologies. There are several reports of using TEM to monitor the recruitment of gold-labelled proteins, such as Ku70 and 53BP1, to DNA lesions generated by high-LET carbon ions [34,54,55]. This provides interesting insight into the spatial localisation of CDD within chromatin and the kinetics of resolving the DNA damage. Despite this, further experimental research using these approaches are needed to fully explore their potential in measuring and monitoring the repair of CDD induced by IR.

### 3.3. Cellular Signalling and Repair of IR-Induced CDD

The activation of the cellular DDR is the most important step in the signalling and processing of damaged DNA, including CDD. Given the nature of CDD which could consist of multiple types of damage including oxidative base damage, SSBs and or DSBs in close proximity, this could require activation of one or a combination of multiple repair pathways including BER, NHEJ and/or HR. This though will very much depend on the nature and degree of complexity of the damage, and which relates to the radiation source and the LET. However, in the literature, there are still conflicting reports as to the specific proteins and pathways involved. When comparing A549 cells irradiated with photons or protons at the centre of a SOBP irradiation, it was found that DNA-PKcs activation was lower in response to protons, and that DNA-PK inhibition sensitised the tumour cells less than that following photon irradiation [56]. It was also found that depletion of RAD51 greatly sensitised the cells to protons suggesting a dependency of HR under these radiation conditions. However, the levels of CDD were not measured directly as being a major contributor to this effect following proton irradiation. The increased utilisation of HR following protons delivered at the Bragg peak, versus entrance dose protons, has also been observed in U2OS cells through the persistence of RPA and RAD51 foci at 4–24 h post-irradiation [57]. Interestingly, it has been shown using 3D spheroids of pancreatic ductal adenocarcinoma (PDAC) that there are significant changes in the phosphoproteome comparing protons at the SOBP and photons, but that inhibition of NHEJ through targeting either ATM or DNA-PK was able to radiosensitise the cell models irrespective of the radiation type [58]. However, again the relationship to CDD was not investigated and confirmed. In contrast, using high-LET CIRT compared to relatively low-LET protons, where the levels of CDD will no doubt differ, DNA repair deficient Chinese Hamster cells were used to demonstrate a preference for HR following CIRT, compared to protons that were thought to largely utilise NHEJ for repair [59]. This is supported by evidence in HeLa and U2OS cells that following high-LET Fe and carbon ions, there are significant increases in DNA end resection involving CtIP at complex DSBs, and which could produce a shift to utilisation of HR [60].

It has been suggested that the production of short DNA fragments following high-LET radiation may contribute to the enhanced biological effect and in fact disrupts NHEJ [61]. This was shown using γ-ray versus neutron-irradiated DNA and monitoring DNA-PK activity, where it was observed that short DNA fragments (<32 bp) may act as direct inhibitors of NHEJ. The inability of Chinese hamster ovary cells to efficiently repair DSBs induced by high-LET Fe or carbon ions through NHEJ [62], plus the observed lack of impact of the radiation on NHEJ-deficient Chinese hamster ovary cells and mouse embryonic fibroblasts versus low-LET X-rays [63], further suggests that NHEJ is inhibited by CDD induced by high-LET radiation. Similarly, the use of engineered clusters of closely spaced DSB sites (~60–200 bp apart) in Chinese hamster ovary cells through the I-SceI nuclease was shown to supress NHEJ relative to isolated DSB sites, but where there was an increased utilisation of aNHEJ for repair that yields a high probability of chromosomal aberrations [33]. High-LET particles can produce spatially correlated CDD as the densely ionising track traverses DNA wrapped around histones forming the nucleosomes and higher order structures such as the chromatin fibre and associated loops. Experimental and theoretical studies have shown an enhancement in DNA fragments of <300 kbp with a peak at ~80 bp [64,65], so this can result in an underestimation of DSB yields. Additionally, sequencing of high-LET α-particle-induced clonal mutations of the HPRT gene have shown that a number of the mutations across the gene were more complicated than a simple deletion event observed for other mutagens, and consistent with clustering of DNA damage along the α-particle track [66].

Findings from our laboratory have clearly demonstrated that CDD is induced in HeLa and head and neck squamous cell carcinoma cells following low energy (relatively high-LET; 12 keV/µm) proton irradiation at the distal end of a SOBP, compared to high energy (relatively low-LET; 1 keV/µm) protons generated at the beam entrance through the use of an enzyme-modified neutral comet assay [13]. Interestingly, we observed that the CDD induced was largely SSB and not DSB-associated, as observed by the delays in SSB repair visualised through the alkaline comet assay. We subsequently discovered using an siRNA screen targeting individual deubiquitylation enzymes to examine those specifically involved in CDD repair, that the ubiquitin-specific protease 6 (USP6) was essential for the survival of cells following high-LET protons through its role in stabilising PARP-1 [27]. Indeed, targeting PARP-1 directly through siRNA or using the inhibitor olaparib was able to mimic the effect of USP6 depletion by causing a delay in CDD repair and enhancing sensitivity of cells to relatively high-LET protons. This corroborated our previous study highlighting an important role for SSB repair proteins, such as PARP-1, in the resolving of CDD-induced by protons at the SOBP distal edge. We also reported that at the chromatin level, ubiquitylation of histone H2B on lysine 120 occurs in head and neck squamous carcinoma and HeLa cells following relatively high-LET protons, which was catalysed by the E3 ubiquitin ligases male-specific lethal 2 homolog (MSL2) and ring finger 20/40 complex (RNF20/40) [13]. An enzyme-modified neutral comet assay was also used to show that in RNF20/40 and MSL2 siRNA depleted cells, that there was a delay in repair of CDD associated with an increased sensitivity of cells to relatively high-LET protons, suggesting that this mechanism is important in CDD recognition and repair at the level of chromatin.

### 3.4. Opportunities for Targeting Enzymes Involved in CDD Repair

DDR inhibition is a revolutionary form of anti-cancer treatment taking advantage of rapidly dividing cancer cells having either a reduced DDR capacity or proficiency. It is therefore possible that DDR inhibitors can selectively kill certain cancer cells whilst sparing normal cells that are DNA repair proficient. Indeed, one of the first successful and well established targeted therapies which utilised this synthetic lethal approach in tumour cells was utilising PARP inhibitors, such as olaparib, in BRCA1/2-deficient breast and ovarian cancers that are unable to efficiently perform HR [67,68]. Synthetic lethality based targeted therapies have a few advantages over conventional anti-cancer therapies, and are particularly less likely to generate treatment resistance caused by conventional therapies, including radiotherapy and chemotherapy alone. Nevertheless, the use of IR and particularly of high-LET radiation that induces significant levels of CDD can be used as a strategy to create synthetic lethality particular in combination with DDR inhibitors. As mentioned previously, the two major protein kinases involved in the activation of the DDR in response to DSBs are ATM and ATR, and along with DNA-PK involved in cNHEJ, these are targets for small molecule inhibitors. Tumour cells also frequently harbour mutations in the p53 tumour suppressor protein that is required for activation of the G_1_/S cell cycle checkpoint allowing time for the cells to repair any DNA damage. Consequently, the checkpoint kinases CHK1 and WEE1 that mediate the G_2_/M checkpoint are also thought to be important therapeutic targets for inhibitors in combination with IR in p53 mutated cells, as this will disable any cell checkpoint activation and increase the likeliness that DNA damage will persist and ultimately lead to cell death. To this effect, we have recently summarised some of the preclinical and clinical evidence supporting key proteins within the cellular DDR as targets for radiosensitisation in head and neck squamous cell carcinoma and glioblastoma [69,70].

In terms of selective radiosensitisation of cells through inhibition of CDD repair, particularly that generated by high-LET IR, the number of reports available to date are scarce. An increasing number of studies have been investigating the potential for targeting proteins within the DDR following proton irradiation, which will generate some degree of increased CDD via the Bragg peak and the distal edge. However, these have not strictly focussed on targeting or confirming CDD as a strategy for radiosensitisation. Indeed, a recent study demonstrated that Bragg peak protons that generated decreased survival (RBE = 1.5–1.6) and persistence of γH2AX/RAD51/RPA foci in U2OS and BT549 cells, compared to entrance dose protons, could be targeted using an ATM inhibitor (AZD0156) to further radiosensitise cells through creating toxic DNA repair intermediates generated by NHEJ [57]. Importantly it was also demonstrated that ATM inhibition enhanced the effectiveness of Bragg peak protons in the suppression of tumour growth in a patient-derived xenograft model. Interestingly, an inhibitor against ATR (AZD6738) or DNA-PK (VX-984) was found not to specifically enhance the radiosensitivity of cells to Bragg peak protons. We also recently reported that an inhibitor of ATM (KU-55933), but also of DNA-PK (KU-57788), could radiosensitise 2D and 3D models of head and neck squamous cell carcinoma cells to protons [71]. However, cells were only irradiated with entrance dose protons and so this is less likely correlated to CDD inhibition, but which nevertheless is a focus for further investigation. In contrast to this study, we have directly demonstrated evidence that PARP-1 protein and activity is essential for the repair of CDD induced by relatively high-LET protons (RBE = 1.7–1.9) in HeLa and head and neck squamous cell carcinoma cells [27]. Utilising the PARP inhibitor olaparib, we showed that cells were radiosensitised only under these conditions generating higher levels of CDD, and not with low-LET protons, and that PARP inhibition led to significant and further persistence of CDD as visualised using the enzyme-modified neutral comet assay. Our more recent evidence has further suggested that inhibiting PARP via olaparib and talazoparib can enhance the radiosensitivity of 3D spheroid models of head and neck squamous cell carcinoma cells to protons [72]. Whilst irradiations were conducted using entrance dose protons, the significantly delayed growth of the spheroids following the combination of the strong PARP trapper talazoparib with proton irradiation suggests a possible LET and therefore CDD-dependent effect. Similar enhanced radiosensitisation of head and neck squamous cell carcinoma cells in response to protons at the SOBP with the alternative PARP inhibitor niraparib has also been observed [73]. Collectively, these data suggest that targeting PARP represents an opportunity to increase the sensitivity of tumour to protons through inhibiting CDD repair, although significantly more studies are required to support this.

Separate from the DDR, histone deacetylases (HDACs) that control chromatin structure and gene expression have been suggested as targets for enhancing tumour radiosensitisation [4]. Relating to CDD, the pan-HDAC inhibitor suberoylanilide hydroxamic acid (SAHA, also known as vorinostat) has been shown to increase sensitisation of paediatric sarcoma cell lines (KHOS-24OS and A-204) to carbon ions, and which was associated with a persistence in γH2AX foci post-irradiation [74]. However, a similar response was seen following low-LET X-rays, suggesting that this may not be CDD-specific. Comparable evidence has been observed following treatment of murine B16F10 melanoma cells with SAHA, where radiosensitisation was observed following either carbon ions or γ-rays [75]. Interestingly, this study revealed selective radiosensitisation of the cells to carbon ions in the presence of romidepsin, a Class I-specific HDAC inhibitor. SAHA has in fact been demonstrated to sensitise A549 cells to γ-rays, protons at the SOBP and also higher LET carbon ions [76]. Intriguingly, there was an associated significant delay in resolving of γH2AX foci in these cells from 1–24 h post-irradiation only following carbon ions, although delays were still seen following low-LET γ-rays. In contrast, a more recent study has shown that SAHA, in addition to two other HDAC inhibitors M344 and PTACH, were unable to significantly radiosensitise A549, U2OS and U87 cells to either protons, or higher-LET carbon or oxygen ions, whereas radiosensitisation in response to γ-rays was observed [77]. This casts doubt on whether HDAC inhibition can lead to enhancement of the biological effectiveness of high-LET radiation, particularly through effects on CDD repair. However, more expansive studies using a range of HDAC inhibitors (e.g., SAHA, valproic acid, romidepsin and mocetinostat) with different specificities are necessary to further investigate this.

## 4. Conclusions and Future Directions

CDD is a signature of IR, and of particular importance to high-LET radiation that generates increased levels and complexity of the damage, which contributes to enhanced biological effectiveness compared to conventional photon radiotherapy. Indeed, the inability of cells to effectively recognise and repair CDD through the various DNA repair mechanisms leads to persistence of the damage, which will more likely trigger cell death. In addition, high-LET radiation will also produce multiple closely spaced sites of CDD along individual radiation tracks, which may not only result in DNA fragments, but their close proximity will also encourage misrepair between sites, resulting in chromosome aberrations. These can include large-scale genome rearrangements through translocations between pairs of chromosomes, the formation of dicentrics and rings, inversions within a chromosome or possibly deletions of chromosome fragments (interstitial or terminal), which, if within critical genes, may lead to an impairment in normal cell function and ultimately result in cell death. High-LET radiation has been shown to be efficient in producing complex chromosomal rearrangements and is often considered a biomarker of exposure to high-LET radiation. CDD is therefore an important factor in the treatment of different cancer types by radiotherapy. Despite this, there are still a number of significant uncertainties in relation to CDD-induced by IR.

The first is that CDD is extremely difficult to measure in cells and in tissues post-irradiation. Historically, and continuing to this day, gel electrophoresis of cellular DNA in the presence of DNA glycosylases is an important method for determining the levels of CDD and the kinetics of repair following irradiation. Analysing DSB markers, such as γH2AX, 53BP1 and RAD51 foci, by immunofluorescence microscopy will also remain an important tool to measure DSB-associated CDD. However, these methods suffer in that they do not provide more detailed information on the complexity and proximity of the DNA lesions present within CDD sites. The use of other alternative microscopy-based techniques, such as AFM and TEM, can be used as quantitative methods to analyse CDD and the proteins recruited to these sites at the nanoscale level, and are an important development in the field. Whilst these are high-resolution approaches, the disadvantages are the high costs and technical difficulty which prevent these from being more mainstream. Nevertheless, studies utilising a combination of methodologies would be interesting in providing information about the biological effects and the yields of IR-induced CDD sites at different cellular levels. The second major uncertainty is the proteins and pathways responsive to CDD. This is a challenge given that the complexity of the damage will no doubt differ based on the radiation source, ion type and dose/dose rate. Evidence suggests an increasing involvement for HR in the repair of CDD particularly that are induced by high-LET Fe and carbon ions. However, other reports following proton irradiation appear to suggest important roles for PARP-1, and potentially other SSB repair proteins, in resolving CDD, which will be different in nature to high-LET-induced damage. Further, more expansive, in vitro studies are vital in exploring the molecular mechanisms which cells from individual tumour types utilise to respond to CDD generated by IR of increasing LET.

Despite the limited mechanistic evidence, there are reports that targeting CDD repair (such as through inhibitors against ATM, PARP and potentially HDACs) can exacerbate the effects of high-LET radiation in promoting tumour cell killing. Therefore, it is possible to utilise this approach for creating synthetic lethality within tumour cells through precision targeted irradiation, particularly via high-LET CIRT, combined with drugs that inhibit CDD repair. Tumour radioresistance is a major problem in cancer therapy, and therefore targeting CDD generated by high-LET radiation could be a strategy to potentially overcome this phenotype and to optimise the therapeutic effect of the radiation. Unquestionably, both in vitro preclinical models and in vivo studies using mouse models would need to be employed to fully explore this potential for the effective utilisation of high-LET radiation in the treatment of specific tumours.

## Figures and Tables

**Figure 1 ijms-24-04920-f001:**
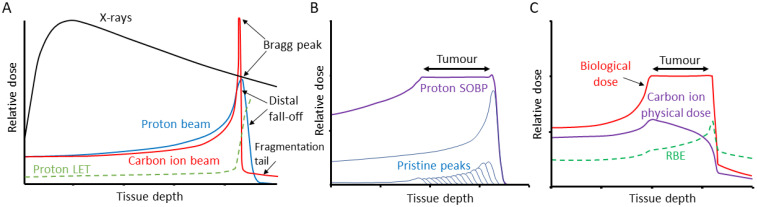
Comparison of tissue penetrance relative to dose of X-rays, PBT and CIRT. (**A**) Tissue-depth-relative dose distribution of X-rays, and pristine proton and carbon ion beams, along with the variation of dose averaged LET for protons varying from ~1 keV/μm to ~17 keV/μm (the LET of carbon ions follows a similar trend, reaching a maximum of ~300–400 keV/μm). (**B**) The use of multiple pristine proton beams of different energies to give a produce a SOBP that generates a relatively uniform dose distribution across a tumour volume. (**C**) The use of multiple carbon ion beams to produce a SOBP, but taking into account the calculated RBE and varying the physical dose to give a relatively uniform biological dose across the tumour.

**Figure 2 ijms-24-04920-f002:**
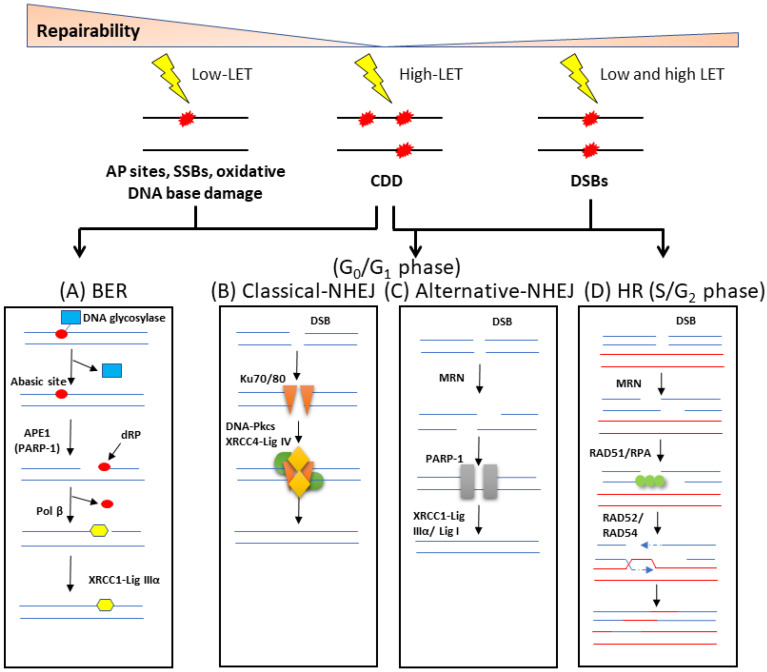
Cellular response to DNA damage by ionising radiation. (**A**) Oxidative DNA base damage, SSBs and abasic (AP) sites are the main types of lesions formed by low-LET radiation, and which are repaired via the BER pathway. In its simplest form, BER is initiated by a damage-specific DNA glycosylase that excises the base damage, the DNA backbone is incised by APE1, the residual 5′-deoxyribosephosphate (dRP) is removed and the single nucleotide gap is filled by Pol β. The SSB remaining is then ligated by the XRCC1-Lig IIIα complex. DSBs created in G_0_/G_1_ of the cell cycle are predominantly repaired by two mechanisms. (**B**) In cNHEJ, Ku70/80 binds to the DNA ends, along with DNA-PKcs and XRCC4-Lig IV to promote DNA ligation. (**C**) In aNHEJ, the MRN complex resects the DNA ends which are then bound by PARP-1, and ligation is performed by XRCC1-Lig IIIα or Lig I. (**D**) During S and G_2_ phases, DSBs can also be repaired by HR which uses a sister chromatid to ensure accurate repair. The MRN complex performs DNA end resection, RAD51 and RPA bind to the single stranded DNA overhangs, followed by strand invasion of the sister chromatid. DNA synthesis, followed by branch migration and Holliday junction resolving is then enabled by RAD52/RAD54. CDD, predominantly generated by high LET radiation, is thought to undergo repair by a combination of these repair pathways, dependent largely on the composition of the damage within the CDD site.

## Data Availability

Data sharing not applicable.

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
