# Peer review of "The Cellular Response to Complex DNA Damage Induced by Ionising Radiation"

_ijms, 2023, doi:10.3390/ijms24054920_

Round 1

Reviewer 1 Report

In the manuscript, the author provides an up-to-date review of the cellular and biological responses to CDD, particularly in the context of IR with increasing LET. Furthermore, the author summarizes some of techniques currently used to measure CDD and ways to target the enzymes and pathways involved in the repair of CDD to enhance the biological effects of IR. This manuscript can be further considered by solving the following questions:

1. Generally, this is a well organized review paper. As HDAC inhibitors were mentioned, drugs targeting related key molecules involving CDD could be added as one part, which will give more clinical relevance and draw abroad readers.

2. The expression of the sentence from line 347 to line 349 When comparing A549 cells irradiated with photons or protons at the centre of a SOBP irradiation, it was found that DNA-PKcs activation was lower in response to protons, and that DNA-PK inhibition sensitised the tumour cells less that that following photon irradiation should be corrected into “… less than that following photon irradiation.

3. It seems that the titles 2.1. Generation and biological consequences of CDD, 2.2. Measurement of CDD, 2.3. Cellular signalling and repair of IR-induced CDD, 2.4. Opportunities for targeting enzymes involved in CDD repair under the title 3. Complex DNA damage (CDD), should be corrected into 3.1. Generation and biological consequences of CDD, 3.2. Measurement of CDD, 3.3. Cellular signalling and repair of IR-induced CDD, 3.4. Opportunities for targeting enzymes involved in CDD repair. Correspondingly, in line 145 is covered in more detail in Section 2.3 should be corrected. Similarly, the title 3. Conclusions and future directions should be corrected into 4. Conclusions and future directions.

4. In the part “2. The Cellular DNA damage response (DDR), The description in the text does not correspond to the content in Figure2. Lines 117-119: NHEJ can be sub-divided into two pathways, namely classical/cannonical NHEJ (cNHEJ) and alternative NHEJ (aNHEJ; Figure 2B). It should be...namely classical/cannonical NHEJ (cNHEJ;  Figure 2B) and alternative NHEJ (aNHEJ; Figure 2C); Line 11, Figure 2C should be Figure 2D.

Reviewer 2 Report

Review of The cellular response to complex DNA damage induced by ionising radiation

I have completed my review of manuscript ijms-2110069, entitled, The cellular response to complex DNA damage induced by ionising radiation.”

About 50% of all human cancers are treated with radiotherapy (ionizing radiation; IR), where the therapeutic effect is primarily attained through DNA damage induction. In particular, complex DNA damage (CDD), which is a hallmark of IR and significantly contributes to the effects of cell killing due to the difficulty of its repair by cellular DNA repair machinery, is defined as having two or more lesions within one to two helical turns of the DNA. The specific DNA repair proteins and pathways, including those involved in DNA single- and double-strand break mechanisms, are uncertain biologically, and CDD repairs heavily depends on the type of radiation and the linear energy transfer that goes along with it (LET). In this context, the topic of the review is important and encouraged.

Comments for authors

Comment 1: I found a lack of explanation in the review article regarding the understanding of how radiation interacts with biological systems and its possible effects. To meet the demand for an introduction section, the suggested article may be useful in this regard. Similar to this article, the inclusion of more literature is recommended for new readers.

Article: Microwave Radiation and the Brain: Mechanisms, Current Status, and Future Prospects. International Journal of Molecular Sciences vol. 23 (2022). [https://doi.org/10.3390/ijms23169288].

Comment 2: In line number 259 The cytokinesis-block micronucleus (CBMN) assay can be used to quantitate micronuclei generated as a result of chromosomal fragmentation. This concept is based on DNA damage, cytostasis, and cytotoxicity in different tissue types, including lymphocytes. If possible, the author could explain a little about its mechanism more how it induces DNA damage could be more efficient for the reader to clearly understand.  

Comment 3: In line number 427 Tumour cells also frequently harbour mutations in the p53 tumour suppressor protein that is required for activation of the G1/S cell cycle checkpoint allowing time for the cells to repair any DNA damage, so the checkpoint kinases CHK1 and WEE1 are also thought to be important therapeutic targets for inhibitors in combination with IR. 

In these lines if author should explain little more mechanism about CHK1 and CHK2 are important as therapeutic drug at which frequency is important to induce these markers so it will use in therapeutic purpose would be more informative to readers.

Comment 4: In line number 501 CDD along individual radiation tracks, which may not only result in DNA fragments, but their close proximity will also encourage misrepair between sites resulting in chromosome aberrations that may impair cell function and ultimately result in cell death.

Here If the authors mention some points about how DNA misrepair induces what are some mechanisms as well as how chromosome aberrations take place because there are several types of chromosome condensation sometimes it will lead to cell mutation also which prepares cells to be more drug-resistant. So, the mechanism about it are more helpful for readers.

Comment 5: For any review, recent advancement is crucial. Discussing recent advancements in the field will strengthen the literature review. I found that the majority of cited articles are older than ten years.

Comment 6: The paper contains errors and typos, I encourage authors to reread carefully and fix any grammatical errors.

Round 2

Reviewer 2 Report

Accept in present form